# Strengthening Mechanisms and Retention Properties of Sintered Iron-Based Matrix Material for Metallic-Diamond Tools

**DOI:** 10.3390/ma16155307

**Published:** 2023-07-28

**Authors:** Elżbieta Cygan-Bączek, Andrzej Romański

**Affiliations:** 1Łukasiewicz Research Network–Krakow Institute of Technology, Zakopiańska 73 Str., 30-418 Krakow, Poland; 2Faculty of Metals Engineering and Industrial Computer Science, AGH University of Science and Technology, 30 Mickiewicz Avenue, 30-059 Krakow, Poland; aromansk@agh.edu.pl

**Keywords:** matrix material, spark plasma sintering (SPS), plastic deformation, deformation twins, grinding, retention

## Abstract

This work presents the analysis of mechanisms controlling the deformation strengthening of the surface during abrasion and their impact on structural changes and mechanical properties of Fe-Mn-Cu-Sn-C matrix material, which was prepared by means of powder metallurgy (PM). The powder mixture was ball-milled for 8 h and densified to <1% porosity using hot pressing at 900 °C and 35 MPa. Phases and structural transformations taking place in austenite during plastic deformation were identified. The participation, distribution, and morphology of the phases, as well as the physicomechanical properties of the matrix material, were tested. It has been shown that during grinding, deformation twins are generated in areas where an austenitic microstructure is present. To test the ability of the matrix to keep diamond crystals firmly cylindrical (Ø11.3 mm× 5 mm), diamond-impregnated specimens containing diamond grits of 30/40 mesh in size and at a concentration of 20 (5% vol.) were prepared. It was finally determined by the diamond-retention index, which is the number of retained diamond particles compared to the total number of diamond particles and the pullouts on the working surface of the segment. This research shows that materials containing Ti- and Si-coated diamond particles, deposited by the CVD method, have the highest abrasion resistance and, therefore, have the best retention properties. In order to determine the bonding mechanism at the matrix–diamond interface, specimens were also analyzed by SEM and TEM.

## 1. Introduction

The stone and construction industry, same as other industries in the world, has been subjected to advancements in technology, which resulted in the development of new and more efficient tools. An increasing demand for metallic-diamond tools, which consist of diamond abrasive grits and a metallic matrix, characterized by greater service life and efficiency in cutting/grinding concrete and rocks, has resulted in the search for new solutions in the field of designing matrix materials with the desired properties. This is mainly due to the fact that metallic-diamond tools work under various conditions and, according to them, both the resistance to abrasive wear and diamond retention of the matrix have to be correctly adjusted to the abrasive properties of the processed material. In natural stone-cutting processes, diamond crystals are subjected to impact loads, which cause a complex state of stress in the matrix material; therefore, it is particularly important to design a matrix material with very good retention and tribological properties. The improvement of the retention properties of the matrix is usually achieved by the use of metalized diamond powders, which, apart from mechanical jamming in the metallic matrix, also form metallurgical connections [1,2,3]. Usually, the highest-quality diamond powders are subjected to metallization using the CVD method. A layer of titanium, chromium, or silicon is applied, with a thickness of 2–10 µm [4]. In addition to increasing retention, the coating protects the diamond crystals from dissolving in the matrix metal and graphitization. Due to the very good retention properties, i.e., the ability to hold diamond particles firmly and the appropriate abrasion resistance, cobalt, and its alloys have been commonly used as a matrix in sintered diamond tools for granite and concrete machining. On the other hand, due to cobalt’s high and very unstable price and, most importantly, its carcinogenicity, from the beginning of the 1990s, intensive research has been undertaken to develop new alloyed powders mainly based on iron and copper [5,6,7,8,9,10,11], which could substitute cobalt and cobalt-base matrices. The current trend in the design of new materials is mainly focused on matrices with increased resistance to abrasive wear [12,13,14,15]. Ratov et al. [16] studied the effect of vanadium nitride additives on the structure and strength characteristics of iron-based matrix composites. Hu et al. [17] investigated the effect of pre-alloyed Fe-based powder amount and consolidation parameters on the performance of impregnated diamond bits. He showed that the pre-alloyed Cr—Fe and pre-alloyed Fe powders can replace tungsten carbide as the framework material of the matrix. Sun et al. [18] developed a new kind of matrix material containing randomly dispersed SiC whiskers. Research has shown that it is possible to obtain a 30% increase in tool life based on SiC-reinforced materials compared to the tools without SiC whisker addition. Levashov et al. [19] investigated the effect of embedded nanoparticles of ZrO_2_, Al_2_O_3_, Si_3_N_4_, and WC on the sintering process and on mechanical and tribological properties of Fe- and Co-based alloys.

An important aspect in the development of matrix materials, apart from ensuring sufficiently high resistance to abrasive wear, is to obtain high retention properties. The improvement of the retention properties of the matrix is usually achieved by the use of coated diamond grits, which, apart from mechanical locking in the metallic matrix, are also bonded in a metallurgical manner [1,2]. Usually, the highest quality diamond grits are coated by a 2–10 µm thick layer of titanium, chromium, or silicon deposited by the CVD method. In addition to increasing retention, the coating protects the diamond crystals from both graphitization and dissolving in the matrix. De Oliveira et al. [20] studied the effect of TiC coatings of diamond crystals, deposited by the CVD method, on the microstructure and mechanical properties of the iron-base composites to compare with composites containing uncoated grits. Zhang et al. [21] evaluated the effect of W-coated diamond fraction on the microstructure and thermal conductivity of the composites. Li et al. [22] investigated the interface and fabrication process of Ti-nanocoated and Cu-nanocoated diamond composites. The nanosized Cu and Ti layers were deposited on the diamond particles in a vacuum ion plating process. The study has shown that the Ti layer deposited on the diamond surface strengthens the interfacial reactions between the Cu- matrix material and nano-sized Ti-layer-coated diamond particles compared to the Cu- matrix material and Cu-coated diamond.

The work Is a continuation of previous research [23,24] regarding a multiphase Fe-Mn-Cu-Sn-C material, which may be a replacement for commercially available materials commonly used in powder metallurgy diamond tools. As shown in [23], this material has a strong strengthening ability during its operation, which results in high resistance to abrasive wear. Compared to Fe-Ni-Cu-Sn-C [7,8] the plastic deformation caused by grinding on the SiC paper had only a limited effect on the amount of deformation-induced martensite, which means that other mechanisms may contribute to strain hardening effect during abrasion. The main purpose of this work was to provide a qualitative and quantitative description of the mechanisms controlling the course of the deformation strengthening of the surface layer of the matrix material under abrasive wear conditions.

## 2. Experimental Method

### 2.1. Materials

The following materials were used to prepare the mixtures: reduced iron powder of NC100.24 grade by Höganäs, Sweden; ground low-carbon ferromanganese powder of XH1218 grade and high-carbon XH1210 grade by ESAB, Poland; and water atomized tin bronze powder containing 20% Sn grade NAM40-80/20 by NEOCHIMIE, France. To obtain the metallic-diamond segments, high-quality uncoated as well as Ti- and Si-coated diamond grits of MBS 970 grade by Hyperion Materials & Technologies, USA/ Columbus and Co- and Cu-coated diamond grits of the GSD 998 grade by CR GEMS Superabrasives, China/Shanghai were used. The microscopic images of the starting powders are presented in Figure 1.

### 2.2. Characterization

The chemical content of the materials are shown in Table 1. The powder mixture was first mixed in a Turbula-type mixer T2C for 1 h and then ball-milled for 8 h in air. The milling container was filled with powder and 12 mm in diameter 100Cr6 steel balls to 50% of its volume and rotated at 70% of critical speed. The ball-to-powder weight ratio was 10:1. The mass fraction of the individual powders was chosen to ensure the following chemical composition of the matrix material: 79.4% Fe 12% Mn 6.4% Cu 1.6% Sn and 0.6% C.

After the milling, the powders were consolidated by Spark Plasma Sintering method in a graphite die at 900 °C for 10 min under 35 MPa. In order to determine the correctness of the consolidation process, the samples were subjected to measurements of density using the Archimedes method and of Rockwell hardness using Future Tech FLC-50VX hardness tester. A total of 10 indentations were made, 5 on each surface perpendicular to the applied compaction force. Specimens were subjected to phase analysis by X-ray diffraction (XRD) using a PANalytical Empyrean diffractometer with Cu radiation (λCu = 1.5406 Å). The obtained data were analyzed by the PANalytical High Score program integrated with the ICDD PDF4 + crystallographic database (2021 year).

In order to determine the behavior of the matrix material (Fe-Mn-Cu-Sn-C alloy) under abrasive wear conditions, laboratory operational tests were performed using RotoPol 21 grinder-polisher by Struers, equipped with a RotoForce-4 head. The tests were carried out simultaneously on three cylindrical samples with a diameter of Ø11.3 mm each, incorporated separately in Struers Epofix resin. The samples were ground on # 220 grit SiC abrasive paper using water as the coolant. During the 20 s measuring cycle, the sample holder was centrally loaded with the force F = 90 N. The specimen holder and the turntable were set to a clockwise motion at 150 and 160 rpm, respectively, which yielded a sliding velocity-altering sinusoid ally between 1.29 and 1.39 m/s. The weight loss per wear interval stabilized the abrasive index A_i2_, being inversely related to the abrasion resistance, was calculated as [25]:(1)$$A_{i2}=\frac{\sum ∆M_{i}}{V\cdot A\cdot (\rho_{s}+2.38)}\cdot 10^{4}\cdot [\mu m/20\cdot m],$$
where $∆M_{i}$ is the mass loss of individual test pieces per 20 s wear interval [g], V is the sliding velocity [m s^−1^], A is the wear surface of the test specimen [cm^2^], and $\rho_{s}$ is the density of the test specimen [g cm^−3^].

After the grinding tests, cross-sections were made and submitted for microscopic examination in order to identify the mechanisms controlling the course of deformation strengthening of the surface layer of the matrix material. The microstructural observations were performed using Versa 3D SEM microscope (FEI, Hillsboro, OR, USA) (field emission electron gun) using the EBSD technique, at an accelerating voltage of 20 kV, and beam current of 20 nA, WD 10 mm. The EBSD Hikari (EDAX) camera was used, and 4 × 4 binning was used, giving 120 × 120 pixel image resolution. The maps were made with a step of 50 nm.

To test the retention properties of the matrix material, metal-diamond segments Ø11.3 mm× 5 mm were made of the 8 h ball-milled Fe-Mn-Cu-Sn-C powders as a matrix and synthetic uncoated and Ti, Si, Co, Cu metalized diamond grits of 30/40 mesh in size and at a concentration of 20 (5% vol.). Diamond-impregnated specimens were tested for wear rate on abrasive concrete using the same Struers equipment. The SiC paper was replaced with C8 grade very abrasive concrete. The test stand with the sample holder is shown in Figure 2. During each 20 s measuring cycle, the segments were individually pressed against the stone abrasive backing wheel with the force F = 20 N. The rotational speed of the backing wheel was 160 rpm. Under the set test conditions, the mean linear speed of the sample was 1.37 m/s.

After each measurement cycle, the segments were cleaned ultrasonically, dried, and weighed individually to the nearest 0.1 mg to calculate the loss of volume. After the grinding test the segments were examined using a stereoscopic microscope (Dino-Lite Digital Microscope AM4013MZT) to determine the number of diamond crystals and pull-out sites on the working surface of the segment.

For microscopic examinations using transmission electron microscopy, thin films were prepared using the FIB ion cannon of the SEM/FIB Quanta 3D 200i (FEI) scanning electron microscope. EDS analyses on selected areas of the foil were performed in the STEM mode at an accelerating voltage of 20 kV.

## 3. Results and Discussion

### 3.1. Powder Characterization

The morphology and microstructure of the particles of the ball-milled powders are shown in Figure 3 whereas the phase composition is presented in Figure 4. The iron and ferromanganese powders were characterized by an irregular shape (Figure 2a–c) and an average particle size in the range of 85–140 µm. The water-atomized tin bronze powder was characterized by a spherical shape and high fragmentation (average particle size 23 µm). During milling, the powder particles are repeatedly flattened, work-hardened, and welded together. The Fe-Mn-Cu-Sn-C alloy powder particles flatten during the milling process. The analysis of the phase composition of the powders has shown that the microstructure consists of the αFe phase, M_23_C_6_-type complex carbides, and the metastable δ phase (Cu_41_Sn_11_).

### 3.2. Composite Characterization

#### 3.2.1. Density and Hardness Measurements

The measurements of the density and hardness of the tested materials were adopted as a criterion for assessing the correctness of the consolidation process. The obtained results are summarized in Table 2. The density values have shown that it was possible to consolidate the tested powder mixtures to a relative density above 98%, which corresponds to an apparent density of 7.75 g/cm^3^. The as-sintered materials are characterized by high hardness > 100 HRB.

#### 3.2.2. X-ray Diffraction

The X-ray diffraction analysis of the composites is presented in Figure 5. According to the Fe−Mn−C [26] phase diagram, it is possible to obtain homogeneous austenite under sintering conditions for the assumed parameters of the consolidation process. The complex carbides containing manganese and iron, present in the ball-milled powder, dissolve in austenite. Rapid cooling after holding at 900 °C hindered the carbides precipitation within the austenite grains. A small amount of carbides can occur at the grain boundaries, which was proven by the CTPi curves of Hadfield cast steel [26] having a similar chemical composition. However, in the case of the investigated material, the carried out diffraction analyses did not indicate any presence of carbides in the microstructure. Obtained data show that the microstructure of the as-consolidated material consists of the following phases: (γFe), (αFe), δ (Cu_41_Sn_11_), and oxides.

#### 3.2.3. Wear Test

As shown in [23], multi-phase specimens from the Fe-Mn-Cu-Sn-C system tend to strengthen in the subsurface layer under the influence of abrasive machining products. As a result of this strengthening, they have high tribological properties, higher than the commercial material used for sintered metallic-diamond tools (Table 2). Laboratory operational tests in the process of grinding on SiC paper were used to deform the surface layer of the tested material, which was then subjected to microscopy tests in order to identify the mechanisms of its strengthening. As shown in [27], a plastically deformed layer is produced both during grinding and polishing. Its depth depends on the sharpness and size of the abrasive, hence the deformed layer stretches to 77 and 0.7 µm after ordinary grinding on SiC # 220 paper and after fine polishing on a 1 µm diamond, respectively.

The obtained results of the A_i2_ abrasive wear resistance tests were referred to the Co-20% WC and Fe-Ni-Cu-Sn reference materials [28], in which the plastic deformation initiates the martensitic transformation in the subsurface layer of the material, creating a hard and wear-resistant martensite.

As shown in Table 2, iron-based materials are characterized by higher abrasive wear resistance despite lower hardness compared to commercial materials.

Confidence intervals were estimated at a 90% confidence level throughout the article.

#### 3.2.4. Microscopic Observation

The LM and SEM analysis of Fe-Mn-Cu-Sn-C material after the consolidated process are presented in Figure 6. After wear tests, the SEM microstructure and the EBSD analysis results are presented in Figure 7 and Figure 8, respectively. As can be observed, as a result of wear, the ferrite regions were deformed (Figure 7a), and twining occurred in the places where the austenitic microstructure was present (Figure 7b). On the worn surface, it is clearly seen that bronze melts during hot pressing and fills all pores. It is also possible that bronze can be melted partially as a result of friction in such conditions (Figure 7c). The EBSD analysis (Figure 8) shows that no martensite was formed in the manganese-rich austenite, as in the case of Fe-Ni-Cu-Sn-C specimens. The misorientation of individual grains is slight, which proves that only the subsurface layer is strengthened by the friction mechanism. No strong deformation in the material was identified. As a result of the plastic deformation of ferrite and austenite grains, the strengthening effect of the material is observed, which improves the abrasive wear resistance of the tested material. In addition, no voids caused by hard particles were found on the worn surface, which proves that they did not crumble from the matrix. No significant changes in the worn surface were reported in the areas where pearlite was present. The depth of the layer affected by the friction process is approximately 30 µm.

#### 3.2.5. Retention Properties

Due to the nature of the work of the metallic-diamond segments, the retention properties of the matrix material are particularly important. During their work, diamond particles transfer variable stresses of different amplitude to the matrix material; therefore, the matrix material should not exhibit any plastic deformation around the embedded diamond grits, which could favor diamond loss. It is therefore essential to ensure good bonding between the matrix and diamond crystals. Most often, the diamond particles are retained in the matrix due to mechanical locking as a result of a large difference in the thermal expansion coefficients of the matrix material and the diamond. This mechanical bonding can be reinforced by a chemical or metallurgical one. To support the mechanical bonding by chemical bond and thus to improve the retention of diamond particles in the matrix, coated diamond grits can be applied. Therefore, diamonds uncoated and coated by Si, Ti, Co, and Cu were used to assess the retention properties of the matrix. The test results are presented in Figure 9, the average number of diamond crystals on the working surface of the segment is given in parentheses.

The gathered results showed that specimens containing Ti- and Si-coated diamond particles have the highest abrasion resistance, so it can be assumed that such a combination of materials provides high retention properties. The samples containing Co- and Cu-coated diamond particles were characterized by the lowest abrasive wear resistance. SEM and TEM examination of the Fe-Mn-Cu-Sn-C + DTi samples showed no additional metallurgical bonding between the matrix and Ti-coated diamond. An example of the analysis of composites containing Ti-coated diamond grains is shown in Figure 10 and Figure 11. From Figure 10, it can be seen that Ti still exists on the surface of the diamond, indicating that Ti-coating was not removed during the fabrication of the specimens. It can also be seen that the diamond particle is wetted by the Cu-Sn liquid phase, which further increases the retention due to the adhesion forces.

In Figure 11, the linear EDS analyses of the chemical composition of the tested sample in selected areas are presented. The remaining analyses confirm the presence of diamond and an austenitic area enriched with manganese and bronze.

## 4. Conclusions

This paper presents the effect of plastic deformation on the change in the structure of Fe-Mn-Cu-Sn-C hot-pressed material subjected to grinding, together with an analysis of the mechanisms controlling the course of deformation strengthening of the surface layer of the matrix material. The specimens were obtained by the SPS method from ball-milled powders. It has been shown that this method can be used to obtain near-pore-free composites. A detailed study of the new material from the Fe-Mn-Cu-Sn-C system, mainly focused on its behavior under abrasive wear conditions, allowed the development of a unique scientific solution that can substitute the commercially available materials mostly based on cobalt. In addition to very good wear resistance, the diamond-retention properties of the investigated materials can be markedly improved by using Ti- or Si-coated diamond grits for segment production in diamond-impregnated tools industry.

## Figures and Tables

**Figure 1 materials-16-05307-f001:**
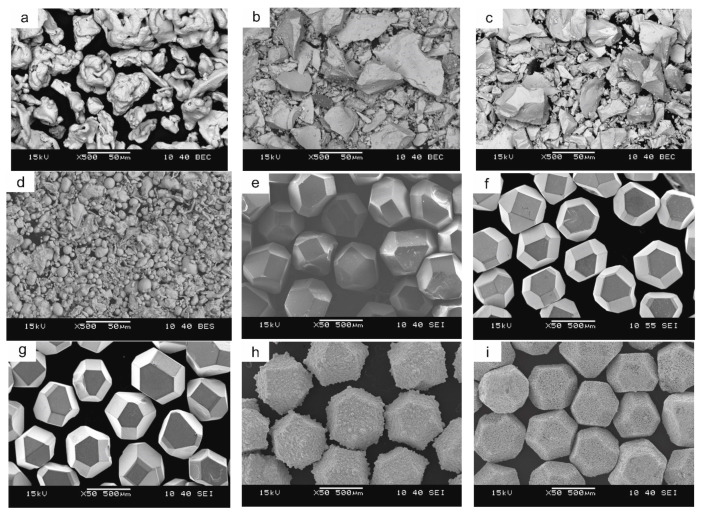
The starting powders: iron grade NC100.24 (**a**), low-carbon ferromanganese grade XH1218 (**b**), high-carbon ferromanganese grade XH1210 (**c**), tin-bronze grade NAM40-80/20 (**d**), uncoated diamond MBS (**e**), and metalized by: Ti (**f**), Si (**g**), Co (**h**) Cu (**i**).

**Figure 2 materials-16-05307-f002:**
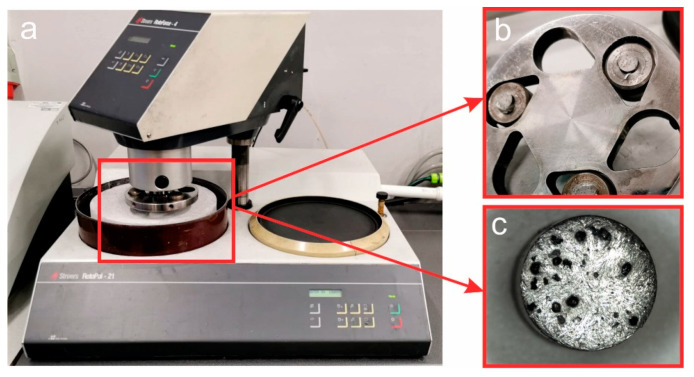
A stand for measuring the wear of metallic diamond segments (**a**), with the sample holder (**b**), and segments (**c**).

**Figure 3 materials-16-05307-f003:**
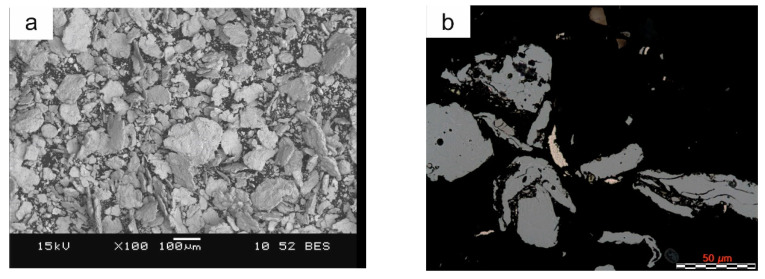
Morphology (**a**) and microstructure (**b**) of powder particles milled in a ball mill for 8 h.

**Figure 4 materials-16-05307-f004:**
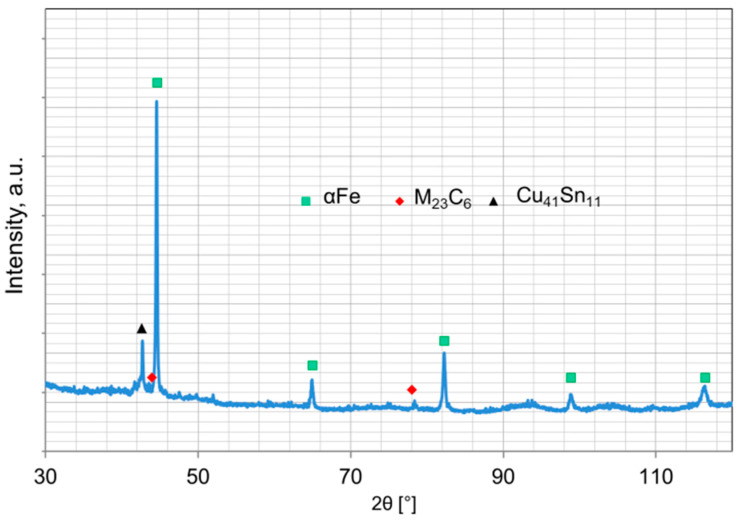
X-ray analysis of a mixture of ground powders for 8 h.

**Figure 5 materials-16-05307-f005:**
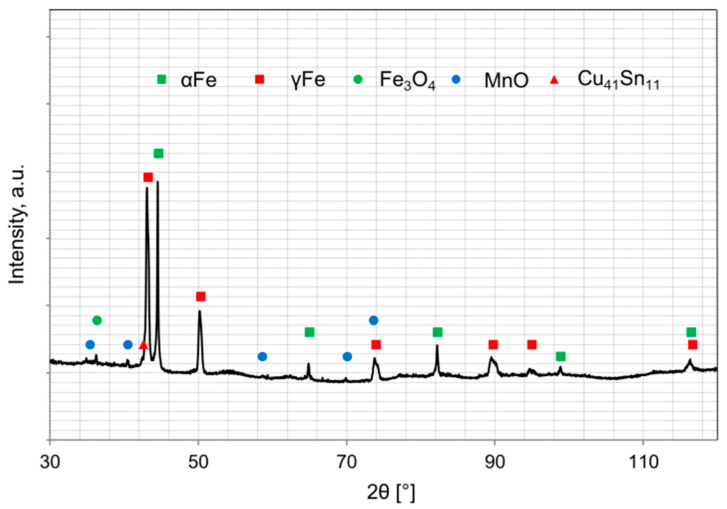
XRD analysis of iron-based composites sintered by SPS at 900 °C.

**Figure 6 materials-16-05307-f006:**
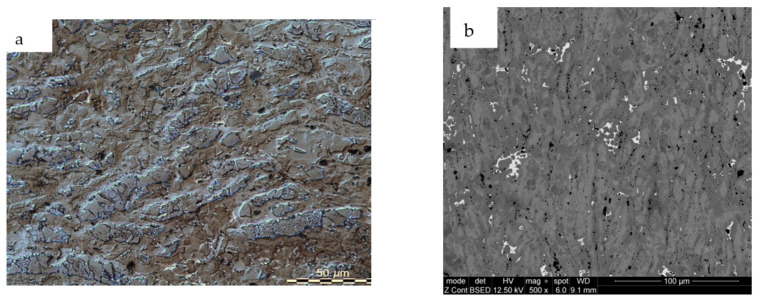
The structure of Fe-Mn-Cu-Sn-C specimens after consolidated using LM (**a**); SEM (**b**).

**Figure 7 materials-16-05307-f007:**
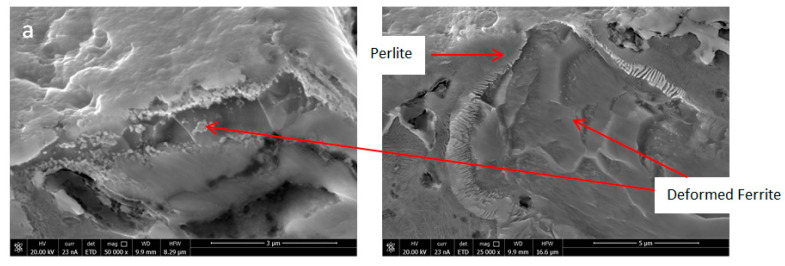
The structure of Fe-Mn-Cu-Sn-C specimens after wear test with visible deformations ferrite grains (**a**), twins (**b**), and bronze on the rubbing surface (**c**).

**Figure 8 materials-16-05307-f008:**
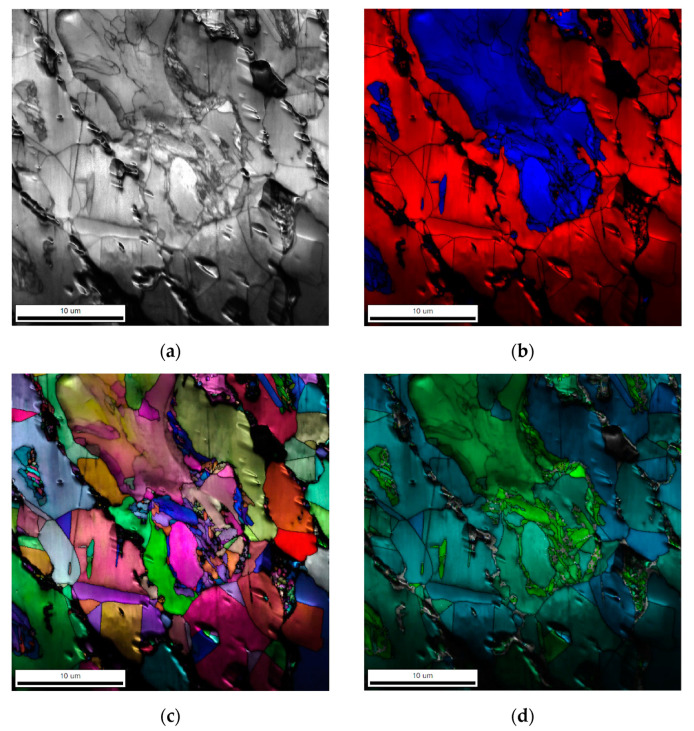
EBSD analysis of the sample after wear test: general view (**a**), phase detection: red—austenite, blue—ferrite (**b**), inverse polar figures (**c**), grain misorientation map (**d**).

**Figure 9 materials-16-05307-f009:**
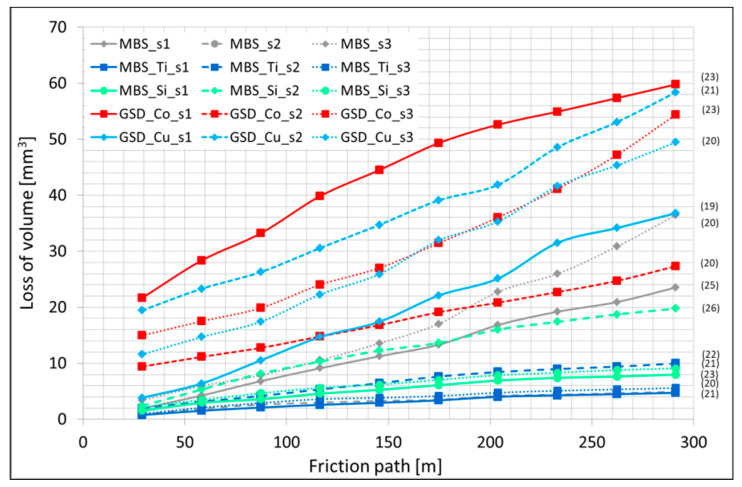
Loss of volume of metallic-diamond segments obtained from ground powder mixtures as a function of the friction path (the average number of diamond crystals on the working surface of the segment is given in parentheses).

**Figure 10 materials-16-05307-f010:**
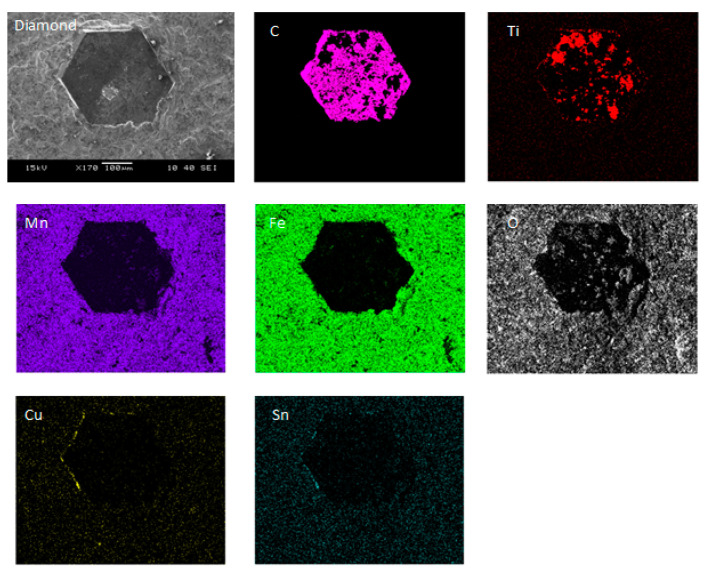
EDS analysis with the distribution of elements on the surface of a Ti-coated diamond.

**Figure 11 materials-16-05307-f011:**
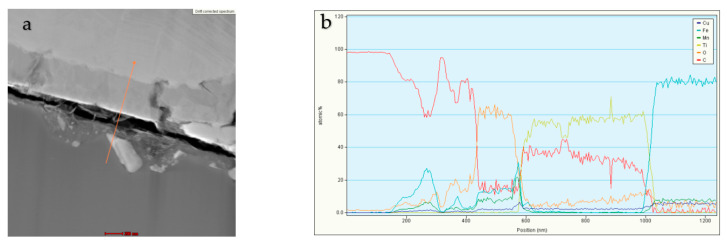
STEM analysis (**a**) with the results of the linear EDS of the sample containing Ti-coated diamond (**b**).

**Table 1 materials-16-05307-t001:** Chemical composition of the starting powders.

Powder	Powder Grade	Chemical Composition, wt.%				
		Fe	Mn	Cu	Sn	C
Iron	NC 100.24	100	…	…	…	…
Ferromanganese	XH1210	Bal	80	…	…	7.0
Ferromanganese	XH1218	Bal	80.5	…	…	1.5
Tin–bronze	NAM40-80/20	…	…	80	20	…

**Table 2 materials-16-05307-t002:** Densities, hardness, and abrasion resistance indices.

Alloy Version	Density^(2)^ [g cm^−3^]	HRB^(2)^	$A_{i2}$ [μm/20·m]
Fe-Mn-Cu-Sn-C	7.75 ± 0.01	103 ± 1	138.7 ± 1.2
Fe-Ni-Cu-Sn-C^(1)^	7.90 ± 0.09	101 ± 2	99.7 ± 5.06
Co-20%WC	9.25 ± 0.01	113 ± 2	177.1 ± 9.2

Data taken from Ref. [28].

## Data Availability

The data presented in this study are available on request from the corresponding author.

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
