# Peer review of "Strengthening Mechanisms and Retention Properties of Sintered Iron-Based Matrix Material for Metallic-Diamond Tools"

_materials, 2023, doi:10.3390/ma16155307_

Round 1

Reviewer 1 Report

Abstract:

-        Abriasion- typo

-        Abstract should highlight the findings of the research – reflecting the title.

Introduction:

-        Line 69- He showed that the use of TiC-coated diamond grits provides better properties compared to composites containing uncoated grits. – He replaced with they.

-        Line 70 – 73. Is there any outcome of the researchers that could relates with the author study?Any improvement on their study?

-        Explain why it is important to study the mechanism–any relationship to final properties?

Method:

-        Figure 1- check label numbered

Results and discussion:

-        Line 232 – ‘As a result of the plastic deformation of ferrite and austenite grains, the strengthening effect of the material is observed’- How the plastic deformation contributed to the strengthening effect?

-        Figure 6 – check label numbering

-        Figure 9 – check labelling

Conclusion:

-        Should contain findings and aims of the study.

-        In the study, is there important to highlight ball milling as they are not highlighted in the introduction as well as relates in discussion e.g assist in strengthening

Minor grammatical error

Author Response

Dear Reviewer,

Thank you very much for you very valuable comments on our manuscript. We have made corrections following suggestions of all reviewers. All changes in the revised version of our manuscript have been highlighted in red.

A detailed reply to your comments, suggestions and remarks has been appended in a separate file (cover letter). We do hope that the current version meets the high publication standards of “Materials”.

Elżbieta Cygan – Bączek

Reviewer 2 Report

(1) In abstract section, authors say;

“The work presents the study of influence of plastic deformation during abriasion on the structural changes and mechanical properties of Fe-Mn-Cu-Sn-C sintered material and the analysis of mechanisms controlling the deformation strengthening of the surface layer.”

Table 1 presents hardness values which data taken from Ref. [26].

However, there is not any graphs or Figure showing the mechanical properties and influence in the manuscript.

(2) The chemical content of the materials produced in the study must be given in a Table in the Materials and Methods section.

(3) The device in Figure 2 is generally used for sanding and polishing. In the current study, it appears to be used for wear tests. However, it is important to indicate its compliance with abrasion test standards and to support its applications with related studies.

(4) As it is known, parameters such as the number of revolutions and the amount of load applied are varied in wear tests. However, in the present study, a single speed (160 rpm) and load (F=20N) were used. It is important to increase the number of tests.

(5) Morphology and microstructure of powders are given in the manuscript. However, it is not seen for sintered material.

Please, provide microstructure and SEM analysis of Fe-Mn-Cu-Sn-C sintered material before wear tests.

(6) Conclusion is too short. It is required to be expanded.

Author Response

(The authors gave the same response as above.)

Reviewer 3 Report

Dear Authors,

It was interesting to review the manuscript You submitted. It concerns the topic that will exist forever. Nevertheless, the article should be easy to read and perceive the information. Please, read the recommendations listed below. I hope they will help You to improve it.

English style should be improved. I have pointed out some moments, but not all. Maybe, the English Editor will help You with this.

When You describe (in the introduction) the coating over the diamond grain, please, name the method by which it was obtained.

Please, explain, why the diamond grains are poorly retained in metal matrices (and give appropriate references in the scientific literature). At least as You did to justify the replacement of cobalt by iron-based alloys

Neither from the summary nor from the introduction, it is unclear, whether You use diamond particles for Your material? What is their size and volume fraction? Were they anyhow coated by metals? By the way, what is the reason to coat the diamond grains before sintering?

The selection of the elements in the metal alloy was not discussed. It just appeared in the article. This can not be so. Please, justify the selection of these elements, and their quantities. What is the expected function of these elements on the alloy? This should be discussed in the introduction.

The main idea and the tasks of the study are not clear. What is of prime importance? If plastic deformation, then the introduction should be focused on it and its effect on material performance. But, if the diamond retention and coating are of prime importance (as it runs from the introduction), then the task of the research should be reformed.

Line 115-116. Did You studied "a material system", or only one alloy containing  79.4% Fe-12% Mn-6.4% Cu-1.6% Sn-0.6% C?

Line 124. Please, in general, explain the measuring method

Line 145-146. The ultrasonic cleaning tends to strike away hard particles embedded in the metal matrix. Did You take this into account?

The abrasion test is very poorly described. Did it really last for 20 seconds only? What was the aim of this test?

Line 206-208. Please, explain the mechanism of deformation, and why the grain size affects the deformation depth. What is the grain size (in micrometers) of the SiC particles on the №220 sandpaper? Put this information into the article. How did You measure the depth of deformation? Did You do any depth hardness profiles to establish that?

Line 228-229: How can we belive that it is mn-reach austenite? Please, provide the corresponding EDS mapping done during EBSD test. Also, please, prove that You have the cementite. Previously, You said that You have M23C6 carbides, and in lines 187-188 You say, that there are no carbides at all (also Fig 5).

Fig. 4: What is the need to do XRD of the mixture? How the carbides were formed? In Fe-Mn-C system, the M23C6 carbide may hardly be formed, The primary one should be (Fe, Mn)3C. You have only 0.6% of C, as in medium carbon steel. I think Fig. 4 may be removed

Figure 5. Where are the diamond particles on the XRD pattern?

Line 233. Where is the microstructure of the material in as-fabricated condition? Where is the microstructure of the material surface before the friction?  How can we estimate the wear test results? Please, give the micrographs of the friction surface with less magnification (200, 500). The diamond grains are big enough to see them.

Line 279-281. This statement is very controversial. Thie maybe happened eventually, particularly in this place. Please, add proof.

Line 287-288. Could it be any other result? You used Ti-coated diamond particles for the material.

English style needs to be revised. The article is readable and understandable but may be significantly improved.

Author Response

(The authors gave the same response as above.)

Round 2

Reviewer 2 Report

The authors have well addressed the comments. There is not any additional comment.

Reviewer 3 Report

Dear Authors,

Thank you for the work done. Now, the manuscript may be published